# '[We] learned how to speak with love': a qualitative exploration of accredited social health activist (ASHA) community health worker experiences of the Mobile Academy refresher training in Rajasthan, India

Kerry Scott [1], Osama Ummer [2,3], Sara Chamberlain [3], Manjula Sharma,[4] Dipanwita Gharai,[2] Bibha Mishra,[2] Namrata Choudhury,[5] Amnesty Elizabeth LeFevre [1,6]

For numbered affiliations see end of article.

**Correspondence to**
Dr Kerry Scott;
kscott26@jhu.edu

## ABSTRACT

**Introduction** Mobile Academy is a mobile-based training course for India's accredited social health activist (ASHA) community health workers (CHW). The course, which ASHAs access by dialling a number from their phones, totals 4 hours of audio content. It consists of 11 chapters, each with their own quiz, and provides a cumulative pass or fail score at the end. This qualitative study of Mobile Academy explores how the programme was accessed and experienced by CHWs, and how they perceive it to have influenced their work.

**Methods** We conducted in-depth interviews (n=25) and focus group discussions (n=5) with ASHAs and other health system actors. Open-ended questions explored ASHA perspectives on Mobile Academy, the course's perceived influence on ASHAs and preferences for future training programmes. After applying a priori codes to the transcripts, we identified emergent themes and grouped them according to our CHW mLearning framework.

**Results** ASHAs reported enjoying Mobile Academy, specifically praising its friendly tone and the ability to repeat content. They, and higher level health systems actors, conceived it to primarily be a test not a training. ASHAs reported that they found the quizzes easy but generally did not consider the course overly simplistic. ASHAs considered Mobile Academy's content to be a useful knowledge refresher but said its primary benefit was in modelling a positive communications approach, which inspired them to adopt a kinder, more 'loving' communication style when speaking to beneficiaries. ASHAs and health system actors wanted follow-on mLearning courses that would continue to compliment but not replace face-to-face training.

**Conclusion** This mLearning programme for CHWs in India was well received by ASHAs across a wide range of education levels and experience. Dial-in audio training has the potential to reinforce topical knowledge and showcase positive ways to communicate.

## STRENGTHS AND LIMITATIONS OF THIS STUDY

⇒ The user perspectives on Mobile Academy presented here are vital to understanding the potential of an mHealth training strategy that requires minimal supervisor input and can be used by CHWs with low or no literacy, limited digital skills and no smartphone or internet access—common realities for CHWs in low-resource settings.

⇒ Our purposive sampling strategy used back-end Mobile Academy data; this sample enabled us to present insight into a variety of user experiences, including perspectives from typical users and a subset of atypically highly educated and digitally savvy ASHAs.

⇒ We are limited to presenting the experiences of ASHAs in the Indian state of Rajasthan; given that Mobile Academy has been scaled up across 13 states, additional localised research is required.

⇒ We present ASHA perspectives on how the programme influenced their knowledge and communication with beneficiaries; evaluating these assertions was beyond the scope of our research.

⇒ The time lag between when ASHAs took Mobile Academy (between 2016 and early 2018) and when we interviewed them about it (late 2018) may have reduced the emotional intensity of their responses or introduced inaccurate recollections of programme components.

## INTRODUCTION

Community health workers (CHWs), a type of front-line health worker (FLHWs), have been widely recognised as an integral part of primary healthcare teams, particularly in low and middle-income countries (LMICs).[1 2] Globally, CHWs perform a range of health system functions to improve the

acceptability and reach of health services, including counselling and health education, encouraging care-seeking and treatment adherence, and improving the reach of healthcare to marginalised communities.[3] However, CHW performance is often hindered by a range of programme and health system limitations, including insufficient training and supervision.[3 4]

With the rapid rise in mobile phone access and network connectivity in LMICs, the use of mobile technology for health (mHealth) has flourished.[5] mHealth strategies have been identified as one mechanism to improve CHW training and support, although evidence of effectiveness is mixed, partially because of the diversity in interventions.[6–8] Digital tools developed specifically for FLHW training range from simple didactic programmes such as informational text messages to complex eLearning courses.[9] WhatsApp groups bolstered supportive supervision for Kenyan CHWs[10] and motivational SMS messages improved the documentation of pregnancies by CHWs in Malawi.[11] However, a human resources management tool called MOTECH did not improve perceived supervision, self-reported motivation, work engagement and job satisfaction among Sierra Leonean CHWs.[12] Research from Bihar, India found that Mobile Kunji, a job aid for accredited social health activist (ASHA) CHWs and *anganwadi* nutrition workers, increased time spent with beneficiaries, improved beneficiary perceptions of health worker trustworthiness and increased some positive health behaviours such as exclusive breast feeding.[13] Continuum of Care Services, a job aid for India's FLHWs, increased the frequency of FLHW–beneficiary interactions and of FLHW coordination of home visits. It also improved job confidence and increased beneficiary practice of skin-to-skin care, immediate breast feeding, and appropriate complementary feeding.[14] It did not improve FLHW supervision or other health-related behaviours among beneficiaries.

User perceptions and acceptability of digital training have broadly been found to be positive. FLHWs have found digital training programmes easy to use and the content easy to understand[15–17] and relevant.[18] However, few studies have conducted in-depth examination of FLHW perspectives on mLearning, including how FLHWs with different profiles engage with the technology and content, motivation and hesitation around a new form of training, appropriateness of content, perceived impact, and how FLHWs compare digital to face-to-face options.

This paper explores user and health system actor perceptions of Mobile Academy, a mobile phone-based interactive voice response (IVR) training programme, which was developed for India's FLHWs, including ASHAs. IVR programmes deliver audio recordings to any type of phone and allows users to provide feedback by pressing a number key on the keypad. Mobile Academy, developed by BBC Media Action in India, provides ASHAs with refresher training on reproductive, maternal, neonatal and child health, and seeks to help them better communicate with community members. The programme consists of 44 short (2.5 min) pre-recorded audio lessons (for a total of 240 min). The ASHAs answer four multiple-choice quiz questions at the end of each of the 11 chapters (for a total of 44 questions) to reinforce the messages and build their confidence, including on using phones for learning. ASHAs dial in to a toll-free number to access the course and can listen to as much content as they like before ending the call. Bookmarking technology allows them to return to the lesson where they left off the next time they call in. At the end of the course, ASHAs who score 50% or higher receive an SMS notifying them of their completion with a unique number; they can show this number to the primary health centre administration to receive a completion certificate. In half of the 13 states where Mobile Academy has been launched, 50%–75% of all ASHAs have successfully completed the course. But in other states, uptake is below 15%.

While the programme has been scaled up by the Government of India and completed by over 180 000 ASHAs to date, knowledge about user perceptions of Mobile Academy, its influence on interpersonal communication and ASHA reflections on future mLearning opportunities remains limited. This paper presents the results of a qualitative study exploring these issues in Rajasthan, a state in northern India.

## METHODS
### ASHA programme and training
ASHAs are female CHWs who facilitate antenatal care, institutional deliveries, and immunisations; provide home-based newborn care; promote family planning, hygiene, and nutrition; and convene village health events.[19 20] Current recruitment standards (released in 2012) dictate that ASHAs are required to have completed 10 years of formal education; ASHAs recruited in the first years of the programme (2005–2012) required only 8 years of education and educational requirements can be 'relaxed' in marginalised regions.[21] While some ASHAs have attained education beyond high school, non-literate ASHAs are not uncommon in many states[22]; 17% of ASHAs surveyed in Madhya Pradesh were unable to read a full sentence,[23] and 30% of ASHAs surveyed in Uttar Pradesh were low literate.[24] ASHAs have been found to average over 20 work hours/week,[25] and receive a small fixed monthly honorarium as well as performance-based remuneration with total monthly income ranging from 900 to 4250 rupees (US$14 to US$65) depending on state-level top-up payments and ASHA activity. ASHAs are also compensated 200 rupees (US$3) for attending face-to-face trainings. ASHAs have long asserted that they are underpaid.[26 27]

State governments aim to provide ASHAs with 23 days of initial training and 10 days of supplementary training, as well as monthly supportive supervision through the government primary healthcare system.[28] Early evaluations identified curriculum inadequacies and low-quality training delivery as major issues facing

the programme.[28 29] Over time, trainings have sought to reorient towards a competency-based approach, with efforts to balance health knowledge, clinical skills, and communication and counselling ability.[30] While the ASHA programme has been credited with bolstering antenatal care, increasing institutional deliveries[31–34] and childhood immunisation,[35 36] and improving community case management of childhood illness,[37–39] studies have consistently identified gaps in ASHA knowledge and skills including on childhood illness and danger signs in pregnancy.[37 39–46] Beyond gaps in knowledge and skills, shortcomings in ASHAs' interpersonal and counselling skills have also been identified.[30 47–49] Their communicative and counselling efficacy is influenced by their training on communication and counselling,[48–50] as well as personal characteristics (eg, the ASHA's confidence, empowerment, education), power hierarchies (eg, caste, class, religion, gender), other identity and relational factors (eg, geographic proximity, political affiliations, marital status, number of children and family relationship histories)[43 51 52] and the health system more broadly (eg, the extent to which ASHAs are able to link to facilities that provide high-quality healthcare).[53 54]

## Study setting

Rajasthan is a majority Hindi-speaking state in northern India with a population of 78 million.[55] In 2015, while 85% of men were literate, literacy among women was only 57%.[56] The 2015/2016 maternal mortality ratio was 199 per 100 000 live births[57] and the rural under-five mortality rate was 54.4 per 1000 live births.[56] Only 39% of women received the recommended four antenatal care check-ups, but 84% of deliveries took place in health facilities.[56] As of January 2018, there were approximately 44 900 ASHAs in the state's rural areas, which is 88% of the targeted recruitment and amounts to 1 ASHA per 1147 rural people.

## Sampling

We received permission to conduct this qualitative study in three districts (Ajmer, Sikar and Pali) of Rajasthan, selected by government authorities. Data collection took place in October and November 2018. ASHAs were selected purposively using data generated by Mobile Academy's technical system and guidance from block-level ASHA facilitators. We interviewed ASHAs who had completed Mobile Academy when it was officially launched (2016, early 2017), as well as ASHAs who completed Mobile Academy more recently (late 2017, 2018). This decision was made to explore possible differences between early and late completers of Mobile Academy. We also hypothesised that late completers would be more likely to recall specific details of the course, since it would be fresher in their minds. We sought ASHAs with varied Mobile Academy engagement profiles. We considered the number of minutes an ASHA was engaged in the course (while most ASHAs in Rajasthan completed the course in approximately 240 min, we sought 11 ASHAs who took

**Table 1** Mobile Academy respondent sample

| Respondent | Male | Female | Total |
|---|---|---|---|
| In-depth interviews (IDIs) | | | |
| ASHA | 0 | 25 | 25 |
| ANM | 0 | 6 | 6 |
| ASHA supervisor | 3 | 3 | 6 |
| Block ASHA coordinator | 2 | 1 | 3 |
| District actor | 3 | 0 | 3 |
| State actor | 1 | 1 | 2 |
| Total | 9 | 36 | 45 |
| Focus group discussions | | | |
| ASHAs (5–6 participants) | 0 | 5 | 5 |

ANM, auxiliary nurse midwife; ASHA, accredited social health activist; IDI, in depth interview.

270 min or more), their scores (while most ASHAs in Rajasthan had a score in the 40s out of a maximum of 44 points, we sampled 15 ASHAs with scores less than 40/44) and ASHAs who had repeated the course (we sampled four ASHAs who went through the course twice). Our focus group discussions (FGDs) included a mix of ASHAs by Mobile Academy completion data, nature of Mobile Academy engagement (quiz scores, etc) and education level. We also interviewed auxiliary nurse midwives (ANMs) and ASHA supervisors, who work closely with ASHAs, and other government health system stakeholders at the block, district and state levels (table 1).

## Recruitment, data collection and ethics

Interviews and FGDs were conducted by four female qualitative researchers (authors MS, DG, BM and NC), supported by a male research manager (OU) and female research coordinator (KS). All researchers were trained over a 1-week period, which included pilot testing the detailed ASHA interview guide, and had a master's level social science education or higher. They approached the respondents first by phone to explain the study, identify themselves as working for a Delhi-based research company, explain that they had governmental approval to conduct this study, and ask if they could meet face to face to learn more and, if the respondents agreed, to participate. The interviews took about an hour and were conducted in ASHA homes and in other stakeholder's offices and health facilities; the FGDs took just over an hour and were conducted in empty school buildings, clinics, and courtyards. ASHA family members were often around the home during the ASHA interviews but did not actively listen to or engage with the interviews. Only the researchers and participants were present during the other data collection. The study information and informed consent was read to each potential participant and then summarised in conversational language to ensure comprehension. All respondents provided informed oral consent. Two people who we approached for the study refused: one ASHA who

## Box 1 Research domains explored

**In-depth interviews**
⇒ Relationship with mobile technology (respondents' access to and use of mobile phones).
⇒ Perception of Mobile Academy (voice, pace, content, relevance).
⇒ Mobile Academy uptake (what motivated ASHAs to complete the course, how they heard about it, technological or social barriers to accessing it).
⇒ Mobile Academy quizzes (difficulty, how respondent felt about them, use of the phone keypad to answer questions).
⇒ Mobile Academy completion certificate (process for receiving certificate, importance of certificate).
⇒ Extent to which Mobile Academy is a refresher training versus providing new content.
⇒ Mobile Academy's influence on ASHA–beneficiary interaction and relationships (listening to Mobile Academy with others, telling others about Mobile Academy, drawing from Mobile Academy to help in communication with beneficiaries; the influence of Mobile Academy on ASHAs' confidence, respect and motivation).
⇒ ASHA face-to-face trainings (content, benefits, drawbacks, relationships among ASHAs and between ASHAs and trainers, link to Mobile Academy).
⇒ Desire for further training, especially through mobile technology (content, modality, dose).

**Focus group discussions**
⇒ General perspectives of Mobile Academy.
⇒ Challenges that ASHAs navigate in trying to promote birth preparedness and family planning.
⇒ Whether Mobile Academy had any influence on how ASHAs spoke to people about these issues.
⇒ ASHA perspectives on future mLearning.

had a family emergency and one state-level actor citing lack of interest. All respondents who agreed to participate in the study also allowed audio recording. The research domains explored are presented in box 1.

### Analysis

Analysis began with daily debrief meetings, where the field team (MS, DG, BM, NC, OU and KS) drew from interview notes to discuss emerging themes and adjust elements of the guide and sampling based on saturation and emergent subtopics without the study domains. After data collection, all audio files were transcribed and translated into English and uploaded into Dedoose, a qualitative data management software. Guided by the principles of thematic analysis, KS and OU developed and applied a coding framework with nine code clusters that echoed the topics explored in the interviews and FGDs. The clusters consisted of between three and 11 codes which were primarily drawn from *a priori* areas of interest.

After coding, we generated code reports that enabled us to read all text tagged by the same code. We present our findings thematically, grouped under the five components of our framework for understanding CHW engagement with mobile learning courses (figure 1).

### Patient and public involvement

The research was shaped by ASHA and other health system actor priorities, experiences and preferences through iterative probing and flexibility within our research domains. Results were disseminated to Government of India stakeholders and actors involved in developing Mobile Academy.

## FINDINGS
### CHW profile

*Mobile Academy was considered relevant and appropriate by ASHAs across a wide range of demographic and engagement profiles*

While the ASHAs in our five focus groups were selected based only on having completed Mobile Academy and being geographically close to one another, the 25 ASHAs selected for the interviews were non-typical in several ways (table 2). We sought the rare ASHAs who completed Mobile Academy late (2018 or late 2017), in a state where 93% of ASHAs completed Mobile Academy in 2016 and early 2017. Our late completers (n=12) turned out to be newly recruited ASHAs who were also more educated than early completers (n=13).[21] This selection skewed our respondent sample towards ASHAs with higher education, which is not representative of ASHAs in the state. Whereas only two (8%) of the ASHAs interviewed in our study had eight or fewer years of education, over 70% of Rajasthan's ASHAs did.[28] However, our five FGDs were composed primarily of 'typical' ASHAs (recruited into the ASHA programme in the mid-2000s, fewer than 10 years of education, completed Mobile Academy early and

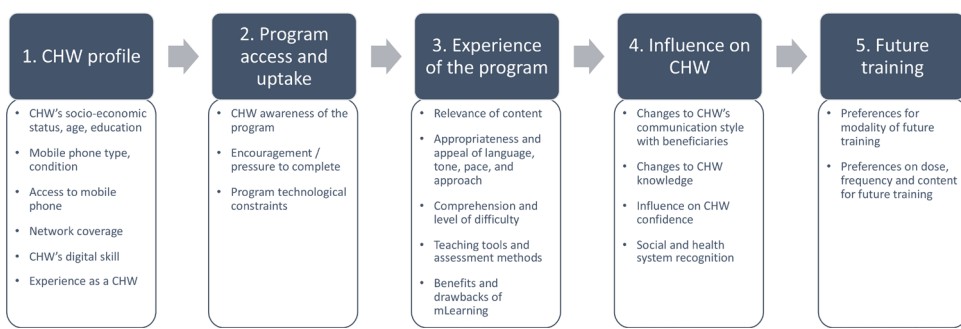

**Figure 1** Framework for understanding community health worker (CHW) engagement with mobile learning courses.

**Table 2** ASHA respondents interviewed in this study compared with ASHAs overall in Rajasthan state

| Parameter | Our IDI sample, n (%) (N=25) | Rajasthan, n (%) (N=see notes) |
|---|---|---|
| Education | | |
| ≤8 | 2 (8%) | 72%* |
| 9–10 | 7 (28%) | 23%* |
| 11–12 | 11 (44%) | 3%* |
| Graduate | 5 (20%) | 1%* |
| Mobile Academy | | |
| Early completion (2016, early 2017) | 13 (52%) | 41 164 (93%)† |
| Late completion (late 2017, 2018) | 12 (48%) | 1440 (3%)† |
| Average quiz score (out of 44 points) | 31.6 | 42.6† |
| Failed (<22/44) | 6 (24%) | 91 (0.2%)† |
| Average duration (min)‡ | 291.2 | 255.0§ |
| ASHAs who repeated the course | 3 (12%) | 533 (13%)§ |

*From *ASHA Which Way Forward* (2011)[28] based on a sample of 200 ASHAs.
†All ASHAs who completed Mobile Academy in Rajasthan (n=44 545).
‡Of first course. If an ASHA repeated the course, this value would approximately double, but we are not considering this second attempt here, since we report repeats in a separate row.
§Only ASHAs who completed Mobile Academy in Sikar, Pali and Ajmer districts.
ASHA, accredited social health activist; IDI, in-depth interview.

achieved high scores), which enabled us to round out the range of perspectives captured in this research.

We did not identify a qualitative difference between early and late completers in terms of their recall of content or their interpretation of Mobile Academy's appropriateness and ease. Both groups reported that the course was a knowledge assessment and a refresher training, which covered the same content as their face-to-face trainings. They reported that the course was relevant and appropriate, and easy to understand. ASHAs across the spectrum of education described the quizzes as easy and enjoyable.

We also sought ASHAs for interviews who had lower scores and who went through Mobile Academy more than once. We included six ASHAs who failed (24% of our sample) when only 0.2% of ASHAs in Rajasthan failed Mobile Academy. Surprisingly, the six ASHAs who failed told us that they are unaware that they had performed poorly—despite the fact that the course informs users when they have failed—and, like the ASHAs with high scores, described the course as 'easy'. All three ASHAs who repeated Mobile Academy (ASHA_11, ASHA_12 and ASHA_24) told us that they did so because their supervisors told them that their first attempt was not properly logged in the system, and that they had to complete the course again. This explanation is likely a miscommunication or misunderstanding because our records show that each attempt was logged.

### ASHAs in our sample had universal mobile phone access and variable digital literacy

All the ASHAs in our sample reported having their own phones, which they carried with them when they went out and could use without asking permission. In many households, the ASHA's phone was also used by other family members. Almost half of the ASHAs in our sample reported that they used a smart phone, which indicates a more digitally enabled group of ASHAs compared with the norm; in Madhya Pradesh a recent survey found that only 11% of ASHAs had a smartphone.[58] Even among our savvy qualitative sample, internet access was variable since few ASHAs bought mobile internet data regularly. (Internet access had no bearing on Mobile Academy access because Mobile Academy is a dial-in mobile phone service.) ASHAs explained that the government-issued Subscriber Identity Module (SIM) card for ASHA-related activities allowed them to speak to other ASHAs and ASHA facilitators and make calls to beneficiaries; however, in some areas, this government SIM card did not work because of network issues or late payment of monthly bills by the government. ASHAs kept a second, personal, SIM card that they sometimes used for ASHA work as well.

Some ASHAs reported receiving calls from government actors, including their ANM and ASHA facilitator, as well as health system actors in district headquarters, the state capital and Delhi. These calls were to inform them about trainings and meetings, ask ASHAs to report data such as the number of pregnancies or, in one case, to ask the ASHA to follow-up with specific high-risk patients. Others had not received any ASHA work-related phone calls. Over half of the ASHAs in this qualitative sample were on WhatsApp, compared with 12% in a representative sample of ASHAs in Madhya Pradesh.[58] Our ASHAs reported using WhatsApp to share reports and information with and among ASHAs; however, access to WhatsApp was irregular and many ASHAs explained that their husbands were actually in the ASHA WhatsApp groups but reported information from the WhatsApp groups back to them.

All ASHAs were confident that they could receive calls from beneficiaries about emergencies, such as women in labour, and could make outgoing calls to the ambulance and to families, often to notify them about vaccinations and health check-ups. Beyond receiving and making calls, most ASHAs self-reported that they could open and read SMS and WhatsApp messages but several disclosed that they got stuck sometimes doing these activities. Our research team did not test these capacities. Most ASHAs said that they were not able to write and send messages by SMS or in WhatsApp, even among those who completed education beyond 12th standard: 'Neither do I know how to send SMS nor do I send. It is not required' (ASHA_08,

44/44, graduate). Several ASHAs described needing help from family members to operate their phones, including saving contacts and opening and reading messages.

## Programme access and uptake
### Mobile Academy was widely understood to be primarily an examination
ASHAs and other health system stakeholders described Mobile Academy as an ASHA knowledge test or examination rather than a refresher course. Relatedly, several higher level stakeholders (DIST_01, BLCK_01, BLCK_02) considered Mobile Academy to be an assessment of the ASHAs and were concerned that Mobile Academy could be completed by other people or that ASHAs could cheat by sharing the answers to the quizzes.

### ASHAs completed Mobile Academy because of pressure from above, but also expressed desire to gain knowledge
ASHAs described pressure to complete Mobile Academy '*aage se* [from above]'. They repeatedly reported being told by their supervisors that completing the course was *zaroori* (necessary/essential) and were also aware that the ANM and ASHA supervisor were under pressure from district-level authorities to have all their ASHAs finish Mobile Academy.

> There was pressure, it's important to do it… we were getting messages, phone calls, again and again, that this is important for you to do. (FGD_ASHA_03)

A few ASHAs described being worried that they would lose their positions if they did not complete Mobile Academy or if they did not pass. One ASHA also reported being told that ASHA pay would increase on completing Mobile Academy, which was not the case (FGD_ASHA_03). ASHAs were not paid for completing Mobile Academy, and when asked about payment they tended to say it was not necessary. Instead, they framed the course as a means to remain in their positions and excel at their work. It was only when discussing future training opportunities that several ASHAs mentioned that they ought to be paid for these activities.

In addition to being required to take the course, ASHAs described being motivated to complete Mobile Academy by their desire to gain more knowledge, as well as general curiosity, an interest in receiving the certificate and a desire to show higher level health system actors that they are knowledgeable.

> We get to learn, that it the reason. There is a will to learn, as an ASHA—for ourselves and for the villagers, like our family, like our village. (ASHA_15, Moble Academy score: 43/44, education level: 12th Standard)

### Mobile Academy uptake among our sample did not follow planned dose and frequency
While ASHAs could call into Mobile Academy and proceed through the course whenever they had time over weeks or even months, the ASHAs we spoke to reported completing it rapidly in just 1 or 2 days. National level data on Mobile Academy engagement found that 36% of ASHAs completed the course within 3 days, and 12% within the first 24 hours.[59] Our data can offer insight into what might have driven this sizeable minority of ASHAs to proceed so rapidly. Some of the ASHAs in our sample reported not knowing that the course could be completed in parts and some were unsure how much time could elapse between starting and completing the course. Others worried about not being able to get through to the Mobile Academy line if they called back later. Many also emphasised that they were under pressure to complete the course quickly.

## Experience of Mobile Academy
### Initial fear replaced by enjoyment; ASHAs praised the pace and tone, valued repetition
When ASHAs first heard that they were to complete a training by mobile phone, they described feeling nervous. They did not know what it would entail or were concerned about having to provide answers to quiz questions. However, they quickly felt comfortable with both the technology and content. ASHAs reported enjoying the course because of the warm and friendly tone, option to repeat content and the fact that the quiz questions covered familiar topics and were easy.

> At first, I was very scared that I would give the wrong answer. Then when I finished it, I felt very good. (FGD_ASHA_05)
>
> I thought I do not know what she will ask and what she will not ask. This was the fear. But after listening I was not afraid. (ASHA_23, 38/44, 10th)

ASHAs expressed very positive overall opinions of the course. The character voices were frequently described as loving, warm and calm: 'Sweetness was there in the voice' (ASHA_24, 44/44, 12th, early adopter). The Hindi language speech was easy to understand for the ASHAs, although they generally spoke Marwadi, a Rajasthani dialect close to Hindi.

> Her voice was good and sweet. […] She spoke exactly like us. (ASHA_29, 35/44, 12th)
>
> We could trust her well. She was talking in such a way that it was nobody else but someone of her own, apna hi hai [like she knows us]. (ASHA_15, 43/44, 12th)
>
> They explained with a lot of love, very slowly. And asked [questions] calmly. (FGD_ASHA_03)

ASHAs felt the pace was appropriate: 'neither it was fast nor slow; everything was explained in a proper manner' (ASHA_29, 35/44, 12th) and greatly valued being able to repeat content. ASHAs emphasised that the repetitive content and ability to relisten to lessons was a major strength of the programme: 'they were repeating some of the topics two to three times and then we were able to remember that nicely' (ASHA_31, 21/44, 8th).

## Quizzes increased ASHA confidence

As designed, the ASHAs experienced quizzes to be easy but enjoyable. Although described as 'easy' by all, ASHAs still said they had to pay attention properly in order to complete them.

> ASHA1: [I] got 100 out of 100. I felt like, if more like these are happening, we will do it.
>
> ASHA2: We had a lot of fun, lot of fun.
>
> ASHA3: That we cleared the paper [passed the test]! If we get the next paper [test], we will clear that too.
>
> ASHA2: We passed it easily. We didn't have to listen to anyone scolding us.
>
> [all laugh]
>
> ASHA4: I felt like I passed a competition.
>
> ASHA3: I was so happy to get good marks. (FGD_ASHA_03)

> Absolutely we have to keep our mind on one point. Whatever they tell we have to find out the answer from that immediately. We must not delay. (ASHA_24, 44/44, 12th)

As mentioned above, those with low quiz scores were unaware that there was any issue with their score and described the quizzes as 'easy'. None reported receiving feedback that they had failed, despite the fact that the course reports back to users whenever they select an incorrect quiz answer and if they have failed at the end, suggesting that ASHAs forgot, did not realise, or did not want to admit to our research team that they had struggled with the quiz.

## Mobile Academy is often a social activity

Despite the individual dial-in design, many ASHAs reported involving other people. Some (ASHA_31, FGD_ASHA_05) reported proceeding through Mobile Academy with a group of other ASHAs during group training sessions. Some played the content for colleagues, such as anganwadi workers and ASHA supervisors, to show them what the course was about or to seek assistance. Others played Mobile Academy on speakerphone for their children or in-laws to explain why they were on the phone for a long duration and to share the content.

## Mobile Academy: a complement to face-to-face training

ASHAs greatly valued that mobile phone-based training allowed them to complete the course from home, when they had spare time. This home-based model avoided the difficulties of making childcare arrangements and leaving behind domestic responsibilities inherent to face-to-face training or meetings.

> Otherwise [for face-to-face training] we have to leave the family and children and go. We have tension in that case. But here we can sit back at home and do the training. We can do it whenever we get time. (ASHA_03, 44/44, 10th)

Furthermore, ASHAs in remote areas valued that mLearning did not require the time and discomfort of travelling long distances to reach training centres. Two ASHAs (ASHA_11, ASHA_18) also noted that they could focus better alone at home compared with group settings where multiple people were speaking at once.

> We can listen to it peacefully. There, [face-to-face training] ten people are sitting. Who is saying what? Here we can listen comfortably. (ASHA_11, 42/44, 8th)

Several ASHAs contrasted the kind and encouraging approach of Mobile Academy with the punitive and demoralising approach to training and supervision that they experienced face-to-face. Some ASHAs reported being frightened of supervisors and trainers, explaining that during some trainings no ASHAs speak because 'we feel a bit scared', 'if we are wrong he will scold us' (FGD_ASHA_03); 'they scold us badly' (FGD_ASHA_05). However, other ASHAs found face-to-face trainers to be supportive and noted that in-person training offered the opportunity to ask questions whenever they did not understand; to learn techniques and practices by seeing them demonstrated; to connect with other ASHAs ('all the ASHAs get together, they talk and gossip' (FGD_ASHA_05)), trainers and facilitators; and to get out of the house.

> ASHA1: Sister, it's like, in training face-to-face we will get to learn more. [...] In Mobile Academy, we can only listen. In the other one, we will get to learn—
>
> ASHA2: The way/technique.
>
> ASHA1: Practical.
>
> ASHA3: They show us.
>
> ASHA2: In face-to-face training, madam, if we ask questions we also get answers. We can ask also, what, why is this. In that [Mobile Academy], we can't even ask. We can just say yes or no. (FGD_ASHA_04)

Most ASHAs explained that both modalities of training had benefits and drawbacks, and thus hoped to receive further face-to-face training, supplemented with mLearning.

## Mobile Academy's influence on ASHAs

### Mobile Academy showcased a positive communication style that ASHAs can apply when speaking to beneficiaries

When asked about Mobile Academy's influence, ASHA most commonly explained that the course changed how they communicated with beneficiaries. They explained that they modelled the tone used by the voice actors in Mobile Academy to speak to beneficiaries, introducing a more loving and calm approach to their communication with women:

> [We] learned how to speak with *prem* [love] (laughs) [...] After doing Mobile Academy, we've made some changes in the way we talk. [...] This only: explain

once. If they don't understand then explain again, but explain with love. Don't get angry, that won't work there. (FGD_ASHA_03)

They spoke of internalising the importance of taking time to talk to beneficiaries, not rushing and repeating content in a caring manner.

### Mobile Academy increased ASHA confidence and reinforced knowledge

ASHAs reported feeling more confident and knowledgeable after completing Mobile Academy. As noted above, Mobile Academy was designed as a refresher training that covered the same topics as face-to-face training; ASHAs reported experiencing it in this way and said that the refresher content enabled them to retain some specific knowledge.

If they tell us again, it makes a difference. […] We are able to understand [now] that the training happened for Mobile Academy. The course refreshes things. [Face-to-face] trainings happened here 8 to 10 years ago. (FGD_ASHA_03)

While ASHAs reported that they did not tell beneficiaries that they completed Mobile Academy and did not listen to Mobile Academy with beneficiaries, they drew from their training, including Mobile Academy, to answer beneficiary questions.

Earlier we did not have courage to talk in front of others. Now we have courage. If we did not know anything, what would we tell others? If we have a little bit of knowledge only then we can tell others. If anybody asks anything, we can answer them as well. (ASHA_15, 43/44, 12th)

While ASHAs were to receive Mobile Academy completion certificates, many did not. ASHAs who had not received a certificate did not articulate personal disappointment or concern about community perception. However, some worried that they lacked proof to show health system stakeholders that they completed the course.

### Future training

ASHAs and higher level health system actors were very positive about integrating future mLearning alongside face-to-face training to reinforce, refresh or deepen understanding. ASHAs wanted future Mobile Academy content to cover newborn care, family planning, nutrition, non-communicable diseases (diabetes, hypertension and cancer), adolescent health, village health committees, tuberculosis, malaria, leprosy and cataracts. They said that future mobile training could consist of up to 6 hours of content, spread in half-hour increments over several weeks or months. When asked directly about incentives to complete future trainings most but not all ASHAs requested payment. One noted that they are paid for face-to-face trainings and said that mLearning ought not to be different: 'We get money for every training. So if we get in this one, it is good' (ASHA_11). However, several said that no money was needed, explaining that the knowledge itself is desirable ('Because we have to learn something, so why should they pay money?' (ASHA_29)) and instead requested other incentives such as certificates or the opportunity for promotion.

## DISCUSSION

This qualitative exploration of Mobile Academy found strong positive perceptions of the course among ASHAs and other health system actors. ASHAs were driven to take the course because of a desire to learn as well as pressure from supervisors. They felt that the warm and friendly tone and their ability to repeat content refreshed their knowledge and showcased a positive communication style that they could apply to interactions with beneficiaries. Respondents were enthusiastic about future mLearning, although they noted that face-to-face training could not be replaced with mobile phone-based programmes. Furthermore, most respondents noted that if mLearning were to become a repeated and regular part of ASHA training, financial incentives would be required to compensate ASHAs for their time and sustain motivation.

In addition to these outcomes, several surprising aspects of ASHA engagement with the course emerged. First, although Mobile Academy was designed for ASHAs with 8 years of education, ASHAs with 12 or more years of education also said that they valued the course. As ASHA educational requirements and mobile phone savvy increase, research examining whether some ASHAs can handle more complex material, alternative teaching styles and the more sophisticated digital engagement used elsewhere will be valuable.[60–62] Nonetheless, given the persistence of low literacy and limited digital skills among women in India[63] including ASHAs,[23 24 64] ongoing investment in audio-based trainings that can be accessed from any phone, rather than textual or smartphone-based programming, remains critical.

Second, although Mobile Academy was designed to be a training programme with simple quiz questions that served only to motivate and engage ASHAs, the quiz component became the focus. Further research is needed to understand the range of effective ways to engage interactive feedback as motivators, teaching tools and systems for assessment and accreditation. Third, the ASHAs in our sample tended to complete the course rapidly, in one or two long sessions, rather than spreading the content over many weeks. Nationally, a smaller portion (36%) took a rapid approach.[59] The ASHAs in our qualitative study explained that they would have liked to proceed through the course more slowly, but took the rapid approach because of pressure from supervisors and concern that they might be unable to access the course again later. This finding, as well as the outsized focus on the quizzes, suggests that supervisors and FLHWs require steady communication about the intent and engagement options for mLearning programmes.

Fourth, ASHAs and health system actors understood Mobile Academy as an internal accreditation rather than an achievement that could bolster community trust. Other studies from South Asia have found that women's engagement with mobile phones enables 'modern' aspirations for education and social mobility but requires reputational risk management.[63 65] The ways in which ASHA use of digital technology affects their social identities and relationships warrant further research.

Power hierarchies in the Indian health system (eg, by education, caste, professional status, gender, urban-rural residence) can be reproduced in trainer-ASHA dynamics and in ASHA–beneficiary interactions, ultimately limiting communication and trust throughout the system.[54 66] ASHAs reported that Mobile Academy's lasting impact on them was both informational and affective—ASHAs spontaneously and frequently focused on the 'loving' communication style and their intention to incorporate this into their interactions with beneficiaries. A few ASHAs contrasted the kind tone in Mobile Academy to a more intimidating teaching style used at some face-to-face trainings. And although many ASHAs expressed sympathy towards families that did not abide by their recommendations, others expressed frustration. ASHAs noted persistent barriers to change, including resistance from mothers-in-law, fear of poor quality care and financial costs at hospitals, and shame and secrecy around reproduction. But several ASHAs described families that did not adhere to their guidance as stubborn, lazy and careless, and felt little shared identity the populations they served. The capacity for digital trainings to improve communication within this hierarchical space requires further exploration: To what extent does Mobile Academy subvert hierarchies versus engage with them in a productive manner? Can face-to-face trainers learn from Mobile Academy's tone and approach? And what is the actual impact of this 'loving' tone on ASHA–beneficiary interactions? Ultimately, it is important to note that strengthening ASHA–beneficiary relationships will require ongoing ASHA capacity building (of which mLearning can play a role), community engagement, and quality improvements in the government health services themselves.[53]

This study was limited to the experiences of ASHAs in the Indian state of Rajasthan. Given that Mobile Academy has been scaled up across 13 states, additional localised research would be beneficial. Another limitation of this study is that we only present ASHA and supervisor perspectives on how Mobile Academy influenced ASHA knowledge and communication with beneficiaries. Evaluating these assertions was beyond the scope of our research but would be very beneficial. Finally, the time lag between when ASHAs took Mobile Academy (between 2016 and early 2018) and when we interviewed them about it (late 2018) may have reduced the emotional intensity of their responses or introduced inaccurate recollections of programme components.

## CONCLUSION

While mLearning programmes have been found capable of improving aspects of FLHW knowledge, patient engagement and supervision, they tend to require smartphone technology and user digital skills that remain out of reach to many of India's rural FLHWs.[38 39] Many also rely on trainer inputs such as remote moderation and feedback,[10] which are challenging add-ons in overstretched systems such as India's where vacancies throughout the supervisory hierarchy remain an issue.[67] Mobile Academy required only a basic phone and the ability to use a keypad to access its pre-recorded content and interactive quiz questions. It was well received by the ASHA FLHWs in our sample, which included a wide range of education levels and experience, including some ASHAs with more education and better digital access than the norm. Policymakers can note Mobile Academy's high acceptability among users, as well as its potential to reinforce knowledge and encourage positive beneficiary–FLHW communication style. These findings position the Mobile Academy model as a viable mLearning option for many LMIC contexts to reinforce ongoing face-to-face training.

**Author affiliations**
¹Department of International Health, Johns Hopkins University Bloomberg School of Public Health, Baltimore, Maryland, USA
²Oxford Policy Management, New Delhi, India
³BBC Media Action, New Delhi, India
⁴Independent researcher, unaffiliated, New Delhi, India
⁵Centre for the Study of Law and Governance, Jawaharlal Nehru University, New Delhi, India
⁶School of Public Health and Family Medicine, University of Cape Town, Rondebosch, Western Cape, South Africa

**Acknowledgements** The authors are grateful to the ASHAs and other health system actors who generously provided their time and insights to make this research possible. This work was made possible by the Bill & Melinda Gates Foundation. We thank Diva Dhar, Suhel Bidani, Rahul Mullick, Dr Suneeta Krishnan, Dr Neeta Goel and Dr Priya Nanda for believing in us and giving us this opportunity. We additionally thank Vinit Pattnaik at OPM and Erica Crawford at Johns Hopkins for their support to the financial management of our work.

**Contributors** KS, SC, OU and AEL conceptualised and designed the study. MS, DG, BM and NC conducted the data collection and preliminary data analysis through daily analytic debriefs. KS and OU led and managed the data collection and analysis, including the coding and thematic synthesis. KS drafted the manuscript and revised it based on critical and substantive input from SC, OU, AEL, MS, BM and NC. All authors agree to be accountable for all aspects of the work related to accuracy and integrity. KS is the guarantor who accepts full responsibility for the work and/or the conduct of the study, had access to the data, and controlled the decision to publish.

**Funding** This work was supported by the Bill & Melinda Gates Foundation (grant number OPP1179252).

**Competing interests** SC is employed by BBC Media Action, one of the entities supporting programme implementation for Mobile Academy.

**Patient and public involvement** Patients and/or the public were not involved in the design, or conduct, or reporting, or dissemination plans of this research.

**Patient consent for publication** Not required.

**Ethics approval** This study involves human participants and was approved by the institutional ethics review boards at Sigma, Delhi, India (10041/IRB/D/17-18) and JHU, Baltimore, USA (00008360). Participants gave informed consent to participate in the study before taking part.

**Provenance and peer review** Not commissioned; externally peer reviewed.

**Data availability statement** Data are available upon reasonable request. Data for this study consist of qualitative interview and focus group discussion transcripts. Uploading all transcripts for open availability would compromise our ability to fully mask participant details. However, we are happy to share anonymised portions of these transcripts upon reasonable request.

**ORCID iDs**
Kerry Scott http://orcid.org/0000-0003-3597-9637
Osama Ummer http://orcid.org/0000-0002-4189-5328
Sara Chamberlain http://orcid.org/0000-0003-4785-6482
Amnesty Elizabeth LeFevre http://orcid.org/0000-0001-8437-7240

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
