## [Reviewer comments · BMJ Open]

ARTICLE DETAILS

TITLE (PROVISIONAL)	"[We] learned how to speak with love": a qualitative exploration of Accredited Social Health Activist (ASHA) community health worker experiences of the Mobile Academy refresher training in Rajasthan, India
AUTHORS	Scott, Kerry; Ummer, Osama; Chamberlain, Sara; Sharma, Manjula; Gharai, Dipanwita; Mishra, Bibha; Choudhury, Namrata; LeFevre, Amnesty

VERSION 1 – REVIEW

REVIEWER	Zabiliute, Emilija The University of Edinburgh
REVIEW RETURNED	20-Apr-2021

GENERAL COMMENTS	This article provides a valuable contribution to the understanding of ASHA learning and training processes and perceptions, training motivations, and engagement with technology. It also provides highly valuable insights into the how technologically-enabled health workers' training is valued and perceived in relationship to other forms of training, and how technologically-enabled learning has unintended effects among ASHAs. The article expands knowledge on the role of technology, affects, learning and communication in Community Health Work and I strongly recommend to publish the article. However, I would like to draw attention to a number of comments and points for revision. Given my own background in cultural anthropology and medical anthropology, the comments and suggestions below relate to conceptual, epistemological and qualitative research questions. • My main comment is related to the research problem, state of the art and the main argument of the paper which would benefit from more precise formulation. At the current stage, the article introduction has an extensive literature review, mainly focusing on the effectiveness of the mLearning among frontline health workers. However, the research problem and its relationship to the article findings remains vague. The article states that it examines ASHAs' "perceptions" of the Mobile Academy. Engaging with literature on ASHA's and CHW's perceptions, everyday and work experiences, and motivations would be useful and perhaps would help to articulate the contributions in a more nuanced way. For instance, is there a relevant bodie(s) of literature that study community health workers' perceptions of training activities or technologies, and what new insights this article brings to such bodies of literature? The abstract claims it contributes to "global understandings of how mLearning programs are experienced by frontline health workers and the potential of future mLearning programs". I would suggest to spell out the contributions in a more specific way.
---

- While the main argument in the introduction could be elucidated, the compelling title of the article seems to suggest its main findings are about the affective value that may ASHAs found in the Mobile Academy's training platform. ASHAs valued a gentle and polite tone and appropriate pace, which, it seems, they found respectful when using the Mobile Academy training. Interestingly, ASHAs compare and contrast this affectionate and likable style of communication to in-person training and encounters with higher-ranking health workers and their supervisors. This points to a curious paradox, where technology is valued for appropriate feelings, kindness and regard, whereas actual human interactions are experienced as inciting fear, anxiety and being unpleasant. Could it be that Mobile Academy here inadvertently subverted the hierarchical, and authoritative atmosphere in India's public health sector? These imbalances can also be linked to gender inequalities, since ASHAs are women, whereas some of the higher-ranking health workers to whom they are accountable, are men. To pursue this direction of inquiry, I would advise to situate these discussions within the literature on user experience, emotions and technology in the context of ASHAs' experiences of regular training and interaction with supervisors.
- Another important contribution this article makes is the finding that the positive and loving" tone of the Mobile Academy influences ASHAs' own style of communication with their communities. As I understand from the article, this is an unintended effect of the Mobile Academy which does not include training on communication style. This relates to the previous comment about affectionate communication and technology. Many qualitative user experience studies emphasize user agency and point out that technology always has unintended consequences and effects among its users. ASHAs valuing the Mobile Academy for affectionate tone would suggest exactly that. Furthermore, the fact that Mobile Academy made and impact on ASHAs' style of communication in their own work also reveals how conversational and communication verbal skills and relationships with communities are important in ASHAs' work along with actual knowledge on medical issues. This speaks to the literature on Community Health Workers' work and caring and affective capacities.
- ASHA payments should be discussed in the article and reflect the terminology used in the article. Accredited Social Health Activists are termed "activists", not workers, because they do not receive regular salaries, differently from other health workers. A vast literature has pointed out to this exploitative aspect of ASHA programme and Community Health Work more generally. Across India, ASHAs, who come from underprivileged background, have been engaging in social protest and mobilisation due to low payments and incentive system. ASHAs receive incentives for different tasks, including for training. As far as I understand from the article, ASHAs are not paid for their Mobile Academy training completion, but use their personal time to complete this training. How does this affect ASHAs' willingness and reception of the training platform, especially in comparison to the regular training sessions, for which ASHAs receive payments? What facets of Mlearning contributed to this positive reception of this training platform despite ASHAs not being paid for their working time?
- The article states that ASHAs have difficulties in interacting with their "beneficiaries". However, the programme design and formal emphasis on the rights-based approach entails that people they engage with are from their own communities – their neighbours and community members. While such difficulties were reported

	before there should be an acknowledgement that ASHAs cultivate more complex relationships in their communities than just ones reducible to a relationship between a “health worker” and “beneficiary”.  • ASHAs being women from underprivileged backgrounds, their work, training and established contacts in governmental public health sector significantly increase their social capital and enable their social mobility. A possible future research could look into whether and how engaging with mobile phone training and other technologies enable women’s aspirations to be socially mobile and “digital” (Quermezi 2017). • Section 5 of the article seems to be very general and depart from the article’s focus on the ASHAs’ learning on Mobile Academy Platform. I would suggest to omit it and instead expand the section on the main argument of the paper, especially discussed in the sections 3 and 4he implications in the discussion section. Alternatively, a link with the main argument should be established. • Literature suggested: Burrell, Jenna. 2011. “User Agency in the Middle Range: Rumors and the Reinvention of the Internet in Accra, Ghana.” Science, Technology, & Human Values 36 (2): 139–59. https://doi.org/10.1177/0162243910366148. Bray, Francesca. 2007. “Gender and Technology.” Annual Review of Anthropology 36: 37–53 Maes, Kenneth 2015 'Volunteers Are Not Paid Because They Are Priceless': Community Health Worker Capacities and Values in an AIDS Treatment Intervention in Urban Ethiopia. Medical Anthropology Quarterly 29 (1): 97–115. doi:10.1111/maq.12136. Mishra, Arima 2014. ‘Trust and Teamwork Matter’: Community Health Workers’ Experiences in Integrated Service Delivery in India. Global Public Health 9 (8): 960–74. Nading, Alex 2013.‘Love Isn’t There in Your Stomach’: A Moral Economy of Medical Citizenship among Nicaraguan Community Health Workers. Medical Anthropology Quarterly 27 (1): 84–102. doi:10.1111/maq.12017. Saprii, Lipekho, Esther Richards, Puni Kokho, and Sally Theobald. 2015 “Community Health Workers in Rural India: Analysing the Opportunities and Challenges Accredited Social Health Activists (ASHAs) Face in Realising Their Multiple Roles.” Human Resources for Health 13 (1): 95. doi:10.1186/s12960-015-0094-3. Swartz, Alison 2013. Legacy, Legitimacy, and Possibility: An Exploration of Community Health Worker Experience across the Generations in Khayelitsha, South Africa. Medical Anthropology Quarterly 27 (2): 139–54. doi: org/10.1111/maq.12020. Swartz, Alison & Christopher J. Colvin (2015) ‘It’s in our veins’: caring natures and material motivations of community health workers in contexts of economic marginalisation, Critical Public Health, 25:2, 139-152, DOI: 10.1080/09581596.2014.941281 Zabiliute, Emilija. Ethics of Neighborly Intimacy among Community Health Activists in Delhi, Medical Anthropology 40:1, 20-34, 2021. Quermezi, Julia Huang 2017. Digital aspirations: ‘wrong-number’ mobile-phone relationships and experimental ethics among women entrepreneurs in rural Bangladesh” Journal of the Royal Anthropological Institute 24, 107-125
--	---

REVIEWER	Olu, Olushayo WHO International
REVIEW RETURNED	19-Jul-2021

GENERAL COMMENTS	This is a very well researched and articulated manuscript on an important topic: use of mobile health technology to improve community health work in India. The introduction is very concise and gives a clear background and strong justification for the study. The qualitative methods used were well described in a such a manner that would facilitate reproduction of the study while the results were also well presented. Although the discussion is brief, it captures most of the findings of the study. I would suggest inclusion of the summary of the key policy recommendations of the study after the conclusion section. What were the limitations of your study and what did you do to mitigate them? Please include a paragraph on this at the end of the discussion section. Otherwise congratulations to the authors for a job well done!
---

VERSION 1 – AUTHOR RESPONSE

Reviewer: 1

Dr. Emilija Zabaliute, The University of Edinburgh

Comments to the Author:

Reviewer 1, comment 1: This article provides a valuable contribution to the understanding of ASHA learning and training processes and perceptions, training motivations, and engagement with technology. It also provides highly valuable insights into the how technologically-enabled health workers' training is valued and perceived in relationship to other forms of training, and how technologically-enabled learning has unintended effects among ASHAs. The article expands knowledge on the role of technology, affects, learning and communication in Community Health Work and I strongly recommend to publish the article.

Response to comment 1: Thank you!

However, I would like to draw attention to a number of comments and points for revision. Given my own background in cultural anthropology and medical anthropology, the comments and suggestions below relate to conceptual, epistemological and qualitative research questions.

Reviewer 1, comment 2: My main comment is related to the research problem, state of the art and the main argument of the paper which would benefit from more precise formulation. At the current stage, the article introduction has an extensive literature review, mainly focusing on the effectiveness of the mLearning among frontline health workers. However, the research problem and its relationship to the article findings remains vague. The article states that it examines ASHAs' "perceptions" of the Mobile Academy. Engaging with literature on ASHA's and CHW's perceptions, everyday and work experiences, and motivations would be useful and perhaps would help to articulate the contributions in a more nuanced way. For instance, is there a relevant bodie(s) of literature that study community health workers' perceptions of training activities or technologies, and what new insights this article brings to such bodies of literature? The abstract claims it contributes to "global understandings of how mLearning programs are experienced by frontline health workers and the potential of future mLearning programs". I would suggest to spell out the contributions in a more specific way.

Response to comment 2: Thank you. Your suggestions are well received. We have edited the introduction to better frame our focus on FLHW perceptions of the program (we do not focus on the actual impact of the program, since assessing changes in knowledge and behavior was outside the scope of our research.) While there is an extensive literature on ASHA motivation and perceptions of their work overall, in light of the word count and need to focus the article on training specifically, we have added relevant literature on CHW perceptions of trainings specifically, with the following text:

"User perceptions and acceptability of digital training have broadly been found to be positive. FLHWs

have found digital training programs easy to use and the content easy to understand^{15–17} and relevant.¹⁸ However, few studies have conducted in-depth examination of FLHW perspectives on mLearning, including how FLHWs with different profiles engage with the technology and content, motivation and hesitation around a new form of training, appropriateness of content, perceived impact, and how they compare them to face-to-face options.”

We have also edited the abstract to be more specific, with the following text in the first paragraph: “This qualitative study of Mobile Academy explores how the program was accessed and experienced by community health workers, and how they perceive it to have influenced their work.”

Reviewer 1, comment 3: While the main argument in the introduction could be elucidated, the compelling title of the article seems to suggest its main findings are about the affective value that may ASHAs found in the Mobile Academy’s training platform. ASHAs valued a gentle and polite tone and appropriate pace, which, it seems, they found respectful when using the Mobile Academy training. Interestingly, ASHAs compare and contrast this affectionate and likable style of communication to in-person training and encounters with higher-ranking health workers and their supervisors. This points to a curious paradox, where technology is valued for appropriate feelings, kindness and regard, whereas actual human interactions are experienced as inciting fear, anxiety and being unpleasant. Could it be that Mobile Academy here inadvertently subverted the hierarchical, and authoritative atmosphere in India’s public health sector? These imbalances can also be linked to gender inequalities, since ASHAs are women, whereas some of the higher-ranking health workers to whom they are accountable, are men. To pursue this direction of inquiry, I would advise to situate these discussions within the literature on user experience, emotions and technology in the context of ASHAs’ experiences of regular training and interaction with supervisors.

Response to comment 3: Thank you for this insight, situating our findings within the power (and gender) hierarchy of the Indian public health sector. We have edited the discussion section as follows:

“ASHAs reported that Mobile Academy’s lasting impact on them was both informational and affective – ASHAs spontaneously and frequently focused on the “loving” communication style and their intention to incorporate this into their interactions with beneficiaries. A few ASHAs contrasted the kind tone in Mobile Academy to a more intimidating teaching style used at some face-to-face trainings. Power hierarchies in the Indian health system (e.g., by education, caste, professional status, gender, urban-rural residence) can be reproduced in trainer-ASHA dynamics and in ASHA-beneficiary interactions, ultimately limiting communication and trust throughout the system.^{54,66} The capacity for digital trainings to improve communication within this hierarchical space require further exploration: to what extent does Mobile Academy subvert hierarchies versus engage with them in a productive manner? Can face-to-face trainers learn from Mobile Academy’s tone and approach? And what is the actual impact of this “loving” tone on ASHA-beneficiary interactions? Ultimately, it is important to note that strengthening ASHA-beneficiary relationships will require ongoing ASHA capacity building (of which mLearning can play a role), community engagement, and quality improvements in the government health services themselves.⁶⁷”

Reviewer 1, comment 4: Another important contribution this article makes is the finding that the positive and loving” tone of the Mobile Academy influences ASHAs’ own style of communication with their communities. As I understand from the article, this is an unintended effect of the Mobile Academy which does not include training on communication style. This relates to the previous comment about affectionate communication and technology. Many qualitative user experience studies emphasize user agency and point out that technology always has unintended consequences and effects among its users. ASHAs valuing the Mobile Academy for affectionate tone would suggest exactly that. Furthermore, the fact that Mobile Academy made an impact on ASHAs’ style of

communication in their own work also reveals how conversational and communication verbal skills and relationships with communities are important in ASHAs' work along with actual knowledge on medical issues. This speaks to the literature on Community Health Workers' work and caring and affective capacities.

Response to comment 4: Yes, thank you for your thoughtful comments on this finding. One thing worth pointing out, is that we don't really know if Mobile Academy influenced their interaction with beneficiaries. We only know that they said it did – we have clearly noted this limitation now in the article discussion.

We have added content in the introduction about training attempts to teach counselling and communication skills to ASHAs, and in the discussion about how Mobile Academy appears to be (according to ASHA self-reports) an mLearning approach that quite powerfully influences these soft skills.

Reviewer 1, comment 5: ASHA payments should be discussed in the article and reflect the terminology used in the article. Accredited Social Health Activists are termed "activists", not workers, because they do not receive regular salaries, differently from other health workers. A vast literature has pointed out to this exploitative aspect of ASHA programme and Community Health Work more generally. Across India, ASHAs, who come from underprivileged background, have been engaging in social protest and mobilisation due to low payments and incentive system. ASHAs receive incentives for different tasks, including for training. As far as I understand from the article, ASHAs are not paid for their Mobile Academy training completion, but use their personal time to complete this training. How does this affect ASHAs' willingness and reception of the training platform, especially in comparison to the regular training sessions, for which ASHAs receive payments? What facets of Mlearning contributed to this positive reception of this training platform despite ASHAs not being paid for their working time?

Response to comment 5: This is a very good question and the ASHA payment issue is major. I do think calling them workers is appropriate since "community health workers" refers to many unpaid (and paid) cadres and because ASHAs receive a form of monthly payment for routine activities, although not a salary.

We have added the following in the section on ASHAs (now in the methods section):

ASHAs have been found to average over 20 work hours a week,²² and receive a small fixed monthly honorarium as well as performance based remuneration with total monthly income ranging from Rs 900 to Rs 4250 (USD\$14 to \$65) depending on state-level top up payments and ASHA activity. ASHAs are also compensated Rs 200 (USD\$3) for attending face to face trainings. ASHAs have long asserted that they are underpaid.^{23,24}

In section 2:

ASHAs were not paid for completing Mobile Academy, and when asked about payment they tended to say it was not necessary. Instead, they framed the course was a means to remain in their positions and excel at their work. It was only when discussing future mLearning opportunities (section 5) that several ASHAs mentioned that they ought to be paid for these activities.

And in section 5:

When asked directly about incentives to complete future trainings most but not all ASHAs requested payment. One noted that they are paid for face-to-face trainings and said that mLearning ought not to be different: "We get money for every training. So if we get in this one, it is good" (ASHA_11). However, several said that no money was needed, explaining that the knowledge itself is desirable ("Because we have to learn something, so why should they pay money?" ASHA_29) and instead requested other incentives such as certificates or the opportunity for promotion.

Reviewer 1, comment 6: The article states that ASHAs have difficulties in interacting with their “beneficiaries”. However, the programme design and formal emphasis on the rights-based approach entails that people they engage with are from their own communities – their neighbours and community members. While such difficulties were reported before there should be an acknowledgement that ASHAs cultivate more complex relationships in their communities than just ones reducible to a relationship between a “health worker” and “beneficiary”.

Response to comment 6: You’re absolutely right. We have added the following text in the introduction “Their communicative and counselling efficacy is influenced by their training on communication and counselling,^{42–44} as well as personal characteristics (e.g., the ASHA’s confidence, empowerment, education), power hierarchies (e.g., caste, class, religion, gender), other identity and relational factors (e.g., geographic proximity, political affiliations, marital status, number of children, and family relationship histories),^{37,45,46} and the health system more broadly (e.g., the extent to which ASHAs are able to linked to clinics that provide high quality health care).^{47,48}”

Reviewer 1, comment 7: ASHAs being women from underprivileged backgrounds, their work, training and established contacts in governmental public health sector significantly increase their social capital and enable their social mobility. A possible future research could look into whether and how engaging with mobile phone training and other technologies enable women’s aspirations to be socially mobile and “digital” (Quermezi 2017).

Response to comment 7: This is a very valuable suggestion. We have added the following to the conclusion:

“Fourth, ASHAs and health system actors understood Mobile Academy as an internal accreditation rather than an achievement that could bolster community trust. Other studies from South Asia have found that women’s engagement with mobile phones enables “modern” aspirations for education and socially mobility but requires reputational risk management.^{57,59} The ways in which ASHA use of digital technology affects their social identities and relationships warrants further research.”

Reviewer 1, comment 8: Section 5 of the article seems to be very general and depart from the article’s focus on the ASHAs’ learning on Mobile Academy Platform. I would suggest to omit it and instead expand the section on the main argument of the paper, especially discussed in the sections 3 and 4 the implications in the discussion section. Alternatively, a link with the main argument should be established.

Response to comment 8: We have now removed this section.

• Literature suggested:

Burrell, Jenna. 2011. “User Agency in the Middle Range: Rumors and the Reinvention of the Internet in Accra, Ghana.” *Science, Technology, & Human Values* 36 (2): 139–59. <https://doi.org/10.1177/0162243910366148>.

Bray, Francesca. 2007. “Gender and Technology.” *Annual Review of Anthropology* 36: 37–53

Maes, Kenneth 2015 ‘Volunteers Are Not Paid Because They Are Priceless’: Community Health Worker Capacities and Values in an AIDS Treatment Intervention in Urban Ethiopia. *Medical Anthropology Quarterly* 29 (1): 97–115. doi:10.1111/maq.12136.

Mishra, Arima 2014. ‘Trust and Teamwork Matter’: Community Health Workers’ Experiences in Integrated Service Delivery in India. *Global Public Health* 9 (8): 960–74.

Nading, Alex 2013. ‘Love Isn’t There in Your Stomach’: A Moral Economy of Medical Citizenship among Nicaraguan Community Health Workers. *Medical Anthropology Quarterly* 27 (1): 84–102. doi:10.1111/maq.12017.

Saprii, Lipekho, Esther Richards, Puni Kokho, and Sally Theobald. 2015 “Community Health Workers in Rural India: Analysing the Opportunities and Challenges Accredited Social Health Activists (ASHAs) Face in Realising Their Multiple Roles.” *Human Resources for Health* 13 (1): 95. doi:10.1186/s12960-015-0094-3.

Swartz, Alison 2013. *Legacy, Legitimacy, and Possibility: An Exploration of Community Health Worker Experience across the Generations in Khayelitsha, South Africa.* *Medical Anthropology Quarterly* 27 (2): 139–54. doi: org/10.1111/maq.12020.

Swartz, Alison & Christopher J. Colvin (2015) ‘It’s in our veins’: caring natures and material motivations of community health workers in contexts of economic marginalisation, *Critical Public Health*, 25:2, 139-152, DOI: 10.1080/09581596.2014.941281

Zabiliute, Emilija. Ethics of Neighborly Intimacy among Community Health Activists in Delhi, *Medical Anthropology* 40:1, 20-34, 2021.

Quermezi, Julia Huang 2017. Digital aspirations: ‘wrong-number’ mobile-phone relationships and experimental ethics among women entrepreneurs in rural Bangladesh” *Journal of the Royal Anthropological Institute* 24, 107-125

Reviewer: 2

Dr. Olushayo Olu, WHO International

Comments to the Author:

Reviewer 2, comment 1: This is a very well researched and articulated manuscript on an important topic: use of mobile health technology to improve community health work in India. The introduction is very concise and gives a clear background and strong justification for the study. The qualitative methods used were well described in a such a manner that would facilitate reproduction of the study while the results were also well presented. Although the discussion is brief, it captures most of the findings of the study. I would suggest inclusion of the summary of the key policy recommendations of the study after the conclusion section.

Response to comment 1: Thank you very much for your positive assessment of our manuscript and methods. We greatly appreciate your review. While we tried to add a paragraph on recommendations, we have found ourselves badly over the allowed word count. We hope that the current summary of findings in the discussion provides adequate directions for policy makers and we have edited the conclusion to synthesize the overall message to policymakers:

“Policymakers can note Mobile Academy’s high acceptability among users, as well as its potential to not only reinforce knowledge but also encourage positive beneficiary-FLHW communication style. These findings position the Mobile Academy model as a viable mLearning option for many LMIC contexts to reinforce ongoing face-to-face training.”

Reviewer 2, comment 2: What were the limitations of your study and what did you do to mitigate them? Please include a paragraph on this at the end of the discussion section.

Response to comment 2: We have added the following paragraph:

“This study was limited to the experiences of ASHAs in the Indian state of Rajasthan. Given that Mobile Academy has been scaled up across 13 states, additional localized research would be beneficial. Another limitation of this study is that we only present ASHA and supervisor perspectives on how Mobile Academy influenced ASHA knowledge and communication with beneficiaries.

Evaluating these assertions was beyond the scope of our research but would be very beneficial. Finally, the time lag between when ASHAs took Mobile Academy (between 2016 and early 2018) and when we interviewed them about it (late 2018) may have reduced the emotional intensity of their responses or introduced inaccurate recollections of program components..”

Reviewer 2, comment 3: Otherwise congratulations to the authors for a job well done!

Response to comment 3: Thank you!

Reviewer: 1

Competing interests of Reviewer: no competing interests

Reviewer: 2

Competing interests of Reviewer: None declared

VERSION 2 – REVIEW

REVIEWER	Zabiliute, Emilija The University of Edinburgh
REVIEW RETURNED	31-Jan-2022

GENERAL COMMENTS	Dear authors and the editor, Thank you once again for this needed and interesting article, and for excellent revisions. I believe the article is ready for publication. I have a few minor comments that would help to finalise it.  • The authors have considered and enhanced and contextualized the debate on ASHA training in relationship to the debates and their preferences on payment. I would suggest to also very briefly – in one sentence - to flag this debate up in the discussion section, for instance, as an important policy recommendation along with the general need for such programmes. If they are used repeatedly and regularly, the incentives for this training would be recommended in order to reward their labour and to sustain motivation and an interest. • As the section five was too general and deleted, I would suggest to add a one-two sentence summary of it into the discussion or conclusion. This could also be an outline of a future possible research on the challenges to implement training knowledge that ASHAs face. • Some of the discussion points could be generalized and linked with each other. For instance, ASHAs felt that Mlearning increased their confidence and knowledge, and that the tone of the learning programme was appropriate. Could this also be that these are linked outcomes, as the acceptable tone and repetitive pedagogy enabled learning and increased their confidence? • I would suggest to rephrase the following sentences/phrases and find generalized descriptions that are more precise and closer to the ASHAs’ responses:  o “Initial fear replaced by happiness” – perhaps satisfaction and being content with a course? o Line 57 – “they quickly relaxed” – they felt comfortable with technology/course content/task....
--

VERSION 2 – AUTHOR RESPONSE

Comment 1: The authors have considered and enhanced and contextualized the debate on ASHA training in relationship to the debates and their preferences on payment. I would suggest to also very briefly – in one sentence - to flag this debate up in the discussion section, for instance, as an important policy recommendation along with the general need for such programmes. If they are used repeatedly and regularly, the incentives for this training would be recommended in order to reward their labour and to sustain motivation and an interest.

Response to comment 1: Thank you for this suggestion. We have added the following as the last sentence of the first paragraph of the discussion: “If mLearning becomes a repeated and regular part of ASHA training, many respondents noted that financial incentives would be required to compensate ASHAs for their time and sustain motivation.”

Comment 2: As the section five was too general and deleted, I would suggest to add a one-two sentence summary of it into the discussion or conclusion. This could also be an outline of a future possible research on the challenges to implement training knowledge that ASHAs face.

Response to comment 2: Great idea. We have added the following to page 15, in the discussion, in the paragraph on hierarchy in the Indian health system: “A few ASHAs contrasted the kind tone in Mobile Academy to a more intimidating teaching style used at some face-to-face trainings. And although many ASHAs expressed sympathy towards families that did not abide by their recommendations, others expressed frustration. ASHAs noted persistent barriers to change, including resistance from mothers-in-law, fear of poor quality care and financial costs at hospitals and shame and secrecy around reproduction. But several ASHAs described families that did not adhere to their guidance as stubborn, lazy and careless, and felt little shared identity the populations they served.”

Comment 3: Some of the discussion points could be generalized and linked with each other. For instance, ASHAs felt that mlearning increased their confidence and knowledge, and that the tone of the learning programme was appropriate. Could this also be that these are linked outcomes, as the acceptable tone and repetitive pedagogy enabled learning and increased their confidence?

Response to comment 3: Thank you. We have edited the first paragraph of the discussion as follows: “They felt that the warm and friendly tone and their ability to repeat content refreshed their knowledge and showcased a positive communication style that they could apply to interactions with beneficiaries.”

Comment 4: I would suggest to rephrase the following sentences/phrases and find generalized descriptions that are more precise and closer to the ASHAs’ responses:

- o “Initial fear replaced by happiness” – perhaps satisfaction and being content with a course?
- o Line 57 – “they quickly relaxed” – they felt comfortable with technology/course content/task....

Response to comment 4: Thank you. We have changed the heading on page 11 to “Initial fear replaced by enjoyment”. The concept of “fear” and “enjoyment” emerged strongly from the data so we thought it would be appropriate to retain these terms. We have changed “quickly relaxed” to “quickly felt comfortable with both the technology and content.”